# Prevalence and risk factors for severe food insecurity and poor food consumption during a drought emergency in Ethiopia

Noah Baker[1], Yunhee Kang[1]*, Gregory Makabila[2], Seifu Tadesse[1,2], Shannon Doocy[1]

1 Johns Hopkins Bloomberg School of Public Health, Baltimore, Maryland, United States of America,
2 Catholic Relief Services, Addis Ababa, Ethiopia

* ykang12@jhu.edu

## Abstract

Frequent drought has heightened nutritional concerns in Ethiopia. This study retrospectively assesses the prevalence and risk factors of severe food insecurity and poor food consumption in Productive Safety Net Programme households in drought-prone Ethiopia. Data was from the USAID-funded Resilience Food Security Activity baseline survey in East Hararghe, Ethiopia. Severe food insecurity (n = 4628; multivariate n = 4335) was defined as Food Insecurity Experience Scale (≥7) and poor food consumption (n = 4554; multivariate n = 4268) was defined as Food Consumption Score (≤21). Logistic regression identified adjusted odds ratio and 95% confidence interval of risk and protective factors. Severe food insecurity prevalence was 77.79% and poor food consumption was 69.74%. Risk factors for severe food insecurity included women/girls aged 15–19 (1.79; 1.36-2.34), current pregnancy (1.51; 1.17-1.96), history of pregnancy (3.46; 2.76-4.33), cash-earning work (1.35; 1.12-1.61), daily-per-capita food consumption <1.61USD (2.98; 1.91-4.66), crop-planting (1.67; 1.31-2.13), and handwashing facilities (3.83; 1.92-7.63); protective factors included two or more children-under-5 (0.72; 0.60-0.88), raising livestock/crops to sell (0.50; 0.42-0.60), and raising oxen (0.34; 0.26-0.45). Risk factors for poor food consumption included woman/girl (1.44; 1.15-1.81) and household-head no education (1.46; 1.18-1.79), daily-per-capita food consumption <1.61USD (4.01; 2.58-6.21), and financial services (2.10; 1.69-2.59); protective factors included women/girls aged 15–19 (0.59; 0.46-0.76) and 30–49 (0.76; 0.63-0.91), two or more children-under-5 (0.77; 0.64-0.91), current pregnancy (0.57; 0.47-0.70), history of pregnancy (0.70; 0.55-0.89), crop-planting (0.57; 0.44-0.75), raising livestock/crops to sell (0.40; 0.34-0.48) and raising oxen (0.68; 0.52-0.90). Vulnerable households included those with pregnant/lactating women, mothers, adolescent girls/women, no education, low assets, and no livestock. Our findings highlight a conceptual distinction, women/girls pregnancy and/or age status may influence household perception and/or definitions of food security despite reporting adequate consumption. The dual burden of food insecurity and poor

**Data availability statement:** The dataset and analytic code used to generate the findings of this study are publicly available on Zenodo at 10.5281/zenodo.15793020. These materials were derived from the baseline survey of the USAID-funded Ifaa Resilience Food Security Activity (RFSA), implemented by a consortium led by Catholic Relief Services ("Causal Design, IMPEL. Baseline Study of the Ifaa Resilience Food Security Activity in Ethiopia. Vol. 1. Washington, DC: The Implementer-Led Evaluation & Learning Associate Award (IMPEL); 2022"). For further inquiries, please contact Noah Baker (nbaker17@alumni.jh.edu) or Yunhee Kang (ykang12@jhu.edu).

**Funding:** United States Agency for International Development- Bureau of Humanitarian Assistance funded the Food for Peace Development Assistance Program: Resilience Food Security Activity (Ifaa project) led by Catholic Relief Services in East Hararghe Zone, Oromia, Ethiopia. Catholic Relief Services was the grant recipient, and the grant number was 720BHA21CA00035. The baseline survey of the Ifaa project was provided to the research team for secondary data analysis. The funders had no role in study design, data collection, data analysis, decision to publish, or preparation of the manuscript.

consumption threatens current and future generations, and data-driven action can help progress towards the goal of zero hunger in Ethiopia.

## Introduction

Ethiopia has increased susceptibility to drought because of its location in the Horn of Africa, proximity to the equator, and proclivity to low rainfall [1]. Climate change has severely impacted the Horn of Africa, with 4 major droughts over the past 15 years, the most recent being the longest recorded drought in decades (2020–2022). Higher drought frequency has led to five consecutive seasons of limited rainfall since 2020 [2]. As of early October 2022, the Horn of Africa was undergoing the worst drought in the past 70 years, with four rainy seasons of below average precipitation [3]. The 2023 Global Humanitarian Overview estimated almost 20 million individuals were affected solely by climactic shocks in eastern and southern Ethiopia, with over 590,000 displaced [4]. The Office for the Coordination of Humanitarian Affairs estimated that 10 million need food assistance and 3 million pregnant/lactating women (PLW) and children need nutritional support. Humanitarian assistance cannot cover these gaps, and in late 2022 the need for agriculture, food, health, and nutrition assistance respectively reached 52%, 82%, 12%, and 48% of the population in drought-affected areas [5].

Historically, the East Hararghe Zone (EHZ) of Oromia has had high levels of household food insecurity [6]. Insufficient rainfall and poor harvests in late 2022 led to emergency levels of food insecurity expected to extend through at minimum mid-2023 [7]. Famine Early Warning System data from November 2020-November 2022 detected that EHZ consistently experiences "crisis" levels of food insecurity, and typically only drop to "stressed" with delivery of humanitarian aid [8]. Many households are reliant on food distribution through humanitarian assistance or social protection schemes such as the Productive Safety Net Programme (PSNP). PSNP households depend on this government assistance program that aims to support the poorest households in Ethiopia, which are at a higher risk for food insecurity [9]. In EHZ, there were upwards of approximately 135,000 internally displaced people, primarily induced by conflict and climate [10]. Displacement can heighten vulnerability to food insecurity and undernutrition, despite many displaced households targeted for food assistance. However, supply chain disruptions, attributed to the war in Tigray and drought, have restricted food assistance and contributed to food price volatility [11]. The Protection Cluster in Ethiopia described high levels of severe food insecurity in EHZ attributed to chronic drought conditions, floods, and pests, which have been compounded by recent conflict [12]. Humanitarian assistance has not been able to cover the nutritional gap and vulnerable populations such as women/girls of reproductive age (WRA) and children under 5 (CU5) are increasingly susceptible to negative nutritional outcomes [8].

Although the Sustainable Development Goals aim to reach zero hunger in Ethiopia by 2030, the 2023 report indicated "major challenges" still remain to achieve this. While progress has been made in child wasting (6.8%), recurrent drought and

conflict, along with socioeconomic barriers, have contributed to the major challenges to reduce the magnitude of under-nutrition (21.9%) and child stunting (36.8%) [13]. With increased drought frequency, high prevalence of food insecurity and low food consumption are likely to remain prevalent in EHZ. The persistent gap between food aid requirements and available assistance primarily experienced by PSNP households reinforces the need to identify a solution. Several studies assess PSNP households, but these primarily evaluated PSNP program effectiveness on food security, food consumption, and asset accumulation [14–17]. Although these studies have demonstrated the positive impacts of PSNP, they maintain limited capacity to provide a large-scale analysis of rural PSNP households in drought emergencies, nor do they address food security and consumption in the context of low availability seasons. There are also no studies that comparatively assess relationships between the consumption-based measure of Food Consumption Score (FCS) and the perception-based measure of Food Insecurity Experience Scale (FIES). Much literature has assessed FCS as a measure of food security, however, the dimensions of food security consist of physical availability, access, utilization, and stability [18], which is difficult to quantify using consumption-based measures that may fluctuate daily. Therefore it is important to analyze this in combination with individual perceptions [19]. The majority of contemporary food security and consumption research in Ethiopia focuses on general populations as opposed to PSNP households, who experience high poverty, lack resources, have limited access to services, and are among the most susceptible in drought emergencies. This analysis examines food insecurity and food consumption of PSNP households to inform sustainable food security and resilience efforts in drought-prone areas. Overall, this study aims to assess the prevalence of and risk factors for severe food insecurity and poor food consumption in vulnerable PSNP households in Ethiopia during a drought emergency.

## Materials and methods

### Data source

This secondary data analysis used data from a cross-sectional survey of PSNP households in eight of the nineteen woredas in EHZ, Oromia, Ethiopia during a drought emergency. The survey was conducted as a baseline for the Ifaa project, a five-year USAID-funded Resilience Food Security Activity implemented by a Catholic Relief Services-led consortium between 2022 and 2026. This study focuses on leveraging existing data to identify potential avenues to improve food security and food consumption in humanitarian emergency settings [20].

Ifaa aims to enhance food security and resilience and targets 4,683 PSNP households with 27,869 beneficiaries. The sample size was based on power calculations and data collection consisted of strict inclusion criteria: households must be PSNP beneficiaries, and one woman/girl of reproductive age (WRA; 15–49 years of age) must reside in the household. Catholic Relief Services collaborated with local authorities and PSNP to enumerate PSNP households with WRA in the target woredas of the Ifaa project. The baseline survey was conducted from May-June 2022 and assessed food insecurity and food consumption respectively in 4678 and 4601 PSNP households in eight EHZ woredas (Babile, Chinaksen, Deder, Fedis, Gursum, Jarso, Melka Belo, and Midega Tola). Of the 241 eligible kebeles, 34 were excluded for security considerations and 11 because of selection for another study, leaving 196 kebeles. A stratified cluster design randomly selected 120 kebeles, and within each kebele, 39 PSNP households with a WRA were randomly selected. This study assessed severe food insecurity in 4,628 households and poor food consumption in 4554 households with descriptive statistics and univariate regression, and for multivariate regression respectively the samples were 4335 and 4268 [20].

The survey respondent was a WRA that resided in the randomly selected household. In cases where multiple WRA resided within the household, the respondent was selected either randomly, or purposively if one WRA was more knowledgeable about the answers to the questions being asked. The selected WRA completed the Women's health, nutritional status, dietary diversity, and family planning modules and provided information on individual and household sociodemographics, economic activity, crop/livestock production, and Water, Sanitation, and Hygiene (WASH). Age groupings indicate the age groups of the WRA respondent. Data was fully anonymized and deidentified before it was provided to the research team. Participants provided oral informed consent. No medical records were included in this research [20].

## Outcomes

**Severe food insecurity (SFI).**  Using the FIES developed by the Food and Agriculture Organization, eight questions were asked to assess the difficulty of households in accessing adequate food in the last 30 days [21]. FIES is an indicator of food insecurity specifically tailored towards data collection in emergency nutrition settings and is commonly used by aid programs and the World Food Programme (WFP) [22,23]. The nature of this measure aligns with this study's objectives to assess food insecurity in drought emergency settings, and with the high caseload of SFI, the need for streamlined data collection tools such as FIES are needed to quickly and accurately assess population food security. Each question consisted of a binary indicator (1 = Yes; 0 = No), and these binary responses were summed to reach a raw score ranging from 0-8. Households were categorized by the FIES scores, with scores ranging from 4-6 indicating moderate food insecurity and 7–8 indicating severe food insecurity [21]. Dummy variables were generated to assess factors associated with the binary outcome of severe food insecurity when FIES ≥7. FIES is used in this study as a summary score to assess the aggregate effects of physical availability, access, utilization, and stability.

**Poor food consumption.**  The household FCS developed by the WFP measures food group consumption frequency, dietary diversity, and nutritional importance [24]. Households were asked how many days a household consumed each of nine food groups in the past seven days: The food groups included: Staples; Pulses; Vegetables; Fruit; Meat/Fish; Milk/Dairy; Sugar/Honey; Oil/Fats; and Condiments [20]. Each food group was weighted by its nutritional value, and these scores were summed ranging from 0-112. Scores ranging from 0-21 indicate poor FCS, 21.5-35 borderline FCS, and ≥35.5 acceptable FCS. A binary variable of poor food consumption (FCS ≤ 21) was generated for further analysis.

## Covariates

Covariates were selected through a literature review and univariate logistic regression. Sociodemographic variables included households with: (1) the WRA respondent age group (constant = 20–29; 15–19; 30–49), (2) number of children under 5 (CU5) in household (none = constant; one; two or more), (3) WRA respondent currently pregnant vs. not pregnant, (4) WRA respondent history of pregnancy vs. no history of pregnancy, (5) household head's no education vs. at least primary education, (6) WRA respondent no education vs. at least primary education, and (7) WRA respondent married vs. not married. Household economic activity and crop/livestock production variables included households that (1) the WRA respondent performed cash-earning work in the past year vs. not performed, (2) saved money vs. did not save, (3) daily per capita food consumption costs <1.61 USD (indicator of below food poverty line based on 2011 purchasing power) vs. daily per capita food consumption costs >1.61 USD (4) used financial services (such as agricultural credit, crop insurance, and/or savings) vs. did not use financial services), (5) had access to a land plot vs. no access to a land plot, (6) raised/purchased livestock and/or cultivated crops with intent to sell vs. did not raise livestock and/or cultivated crops with intent to sell, (7) raised oxen, poultry, or goats vs. did not raise oxen, poultry, or goats, and (8) planted crops they make decisions over vs. did not plant crops they make decisions over. WASH materials/practices included households that (1) had handwashing facilities within the home vs. did not have handwashing facilities in the home, (2) correctly treated water vs. did not correctly treat water, and (3) used household-level improved sanitation facilities (not shared) vs. did not use household-level improved sanitation [20].

## Statistical analysis

Exploratory data analysis was conducted to present n (%) of categorical variables for outcomes (Table 1) and risk factor variables (Table 2). With its larger sample size, descriptive analysis was conducted on the SFI sample as the two-sample z-test for proportions indicated no differences in the distribution of risk/protective factors (Table 2). Inclusion criteria consisted of PSNP households with a WRA respondent 15–49 years of age. Logistic regression was used to identify risk factors significantly associated with the dependent variables (SFI and poor FCS). First, univariate logistic regression

**Table 1. Prevalence of severe food insecurity and poor food consumption score in East Hararghe Zone, Oromia, Ethiopia (Fig 1).**

| Outcome | Frequency | Total Sample | Prevalence |
|---|---|---|---|
| Severe Food Insecurity | 3,600 | 4,628 | 77.79% |
| Poor Food Consumption | 3,176 | 4,554 | 69.74% |

**Table 2. Variable proportions and two-sample z-test for proportions of sociodemographics, economic activity, crop/livestock production, and WASH in severe food insecurity and poor food consumption score samples.**

| Characteristics | Severe Food Insecurity | | | Poor Food Consumption | | | Two-sample z-test for proportions |
|---|---|---|---|---|---|---|---|
| | N | n | % | N | n | % | p-value |
| **Age Group** | 4,628 | | | 4,554 | | | |
| 20–29 y | | 1,085 | | 23.44% | 1,067 | 23.43% | 0.987 |
| 15–19 y | | 757 | | 16.36% | 743 | 16.32% | 0.957 |
| 30–49 y | | 2,786 | | 60.20% | 2,744 | 60.25% | 0.956 |
| **Number of CU5 in household** | 4,628 | | | 4,554 | | | |
| No CU5 in household | | 2,210 | | 47.75% | 2,183 | 47.94% | 0.861 |
| One CU5 in household | | 1,215 | | 26.25% | 1,194 | 26.22% | 0.970 |
| Two or more CU5 in household | | 1,203 | | 25.99% | 1,177 | 25.85% | 0.871 |
| **Demographics** | | | | | | | |
| Currently pregnant | 4,627 | 620 | 13.40% | 4,553 | 606 | 13.31% | 0.899 |
| History of pregnancy | 4,626 | 4,046 | 87.46% | 4,552 | 3,976 | 87.35% | 0.867 |
| Married | 4,628 | 3,069 | 66.31% | 4,554 | 3,023 | 66.38% | 0.945 |
| **Education Level** | | | | | | | |
| WRA No school | 4,628 | 3,981 | 86.02% | 4554 | 3,917 | 86.01% | 0.992 |
| Household head No school | 4,628 | 4,003 | 86.50% | 4554 | 3,937 | 86.45% | 0.951 |
| **Economic activity** | | | | | | | |
| WRA cash earning work last 12 month | 4,628 | 1,332 | 28.78% | 4,554 | 1,291 | 28.35% | 0.646 |
| Household saved money | 4,395 | 120 | 2.73% | 4,326 | 114 | 2.64% | 0.783 |
| Daily per capita food consumption costs <1.61 USD | 4,622 | 4,506 | 97.49% | 4,548 | 4,433 | 97.47% | 0.954 |
| Used financial services | 4,398 | 811 | 18.44% | 4,329 | 793 | 18.32% | 0.883 |
| Access to a plot of land | 4,485 | 4,353 | 97.06% | 4,415 | 4,285 | 97.06% | 0.997 |
| Household raises livestock/crops with intent to sell | 4,396 | 1,006 | 22.88% | 4,328 | 982 | 22.69% | 0.828 |
| **Livestock/Crops** | | | | | | | |
| Household raises oxen | 4,485 | 285 | 6.35% | 4,415 | 279 | 6.32% | 0.946 |
| Household raises poultry | 4,485 | 882 | 19.67% | 4,415 | 871 | 19.73% | 0.941 |
| Household raises goats | 4,485 | 1,052 | 23.46% | 4,415 | 1,035 | 23.44% | 0.988 |
| Household planted crops it made decisions over | 4,351 | 3,967 | 91.17% | 4,283 | 3,907 | 91.22% | 0.939 |
| **WASH** | | | | | | | |
| Household had handwashing facilities | 4,626 | 109 | 2.36% | 4,552 | 109 | 2.39% | 0.904 |
| Household correctly treats water | 4,621 | 342 | 7.40% | 4,547 | 337 | 7.41% | 0.985 |
| Household used improved sanitation facilities | 4,626 | 733 | 15.85% | 4,552 | 726 | 15.95% | 0.892 |

identified variables significant at the 95% confidence level. Variables significant at the 95% confidence level in univariate regressions were retained for the multivariate model. Covariates significant at the 95% confidence level (α = 0.05) in multivariate models were considered risk/protective factors of SFI and poor FCS. Complete case analysis was utilized and respondents with missing data were included in the descriptive and univariate regression, but excluded from the

multivariate regression. Covariates that displayed collinearity were excluded. Stata 17 (STATA Co.) was used for the statistical analysis.

## Ethical approval

Causal Design attained Institutional Review Board (IRB) approval from the Ethiopian Society of Sociologists, Social Workers, and Anthropologists prior to Ifaa baseline data collection [20]. The study authors received an exempt determination from the Johns Hopkins Bloomberg School of Public Health IRB (FWA: #00000287).

## Informed consent

Participants received informed consent during the first module of the Ifaa baseline questionnaire. Participants were informed on the survey purposes, topics, and the person's rights as a participant, including that their responses will remain confidential and that participation was voluntary. Contact information for the investigators and IRB information was provided. Data collectors conducted the survey in the local language (Oromo) and translated data. Oral consent was provided by participants [20].

## Results

Out of 4,628 households, the prevalence of severe food insecurity in this sample was very high at 77.79%, and out of 4,554 households, most (69.74%) reported poor food consumption (Table 1) (Fig 1). The distribution of age groups indicated that 60.20% of the household's WRA respondents were 30–49 years, 16.36% were 15–19 years, and 23.44% were

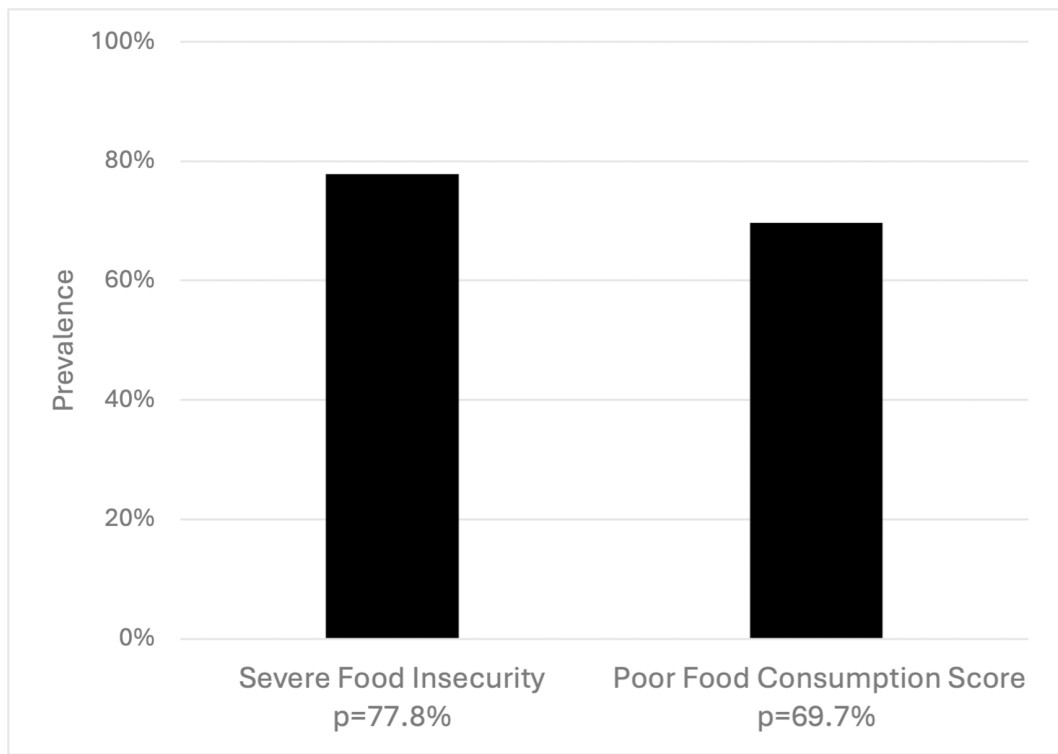

**Fig 1. Prevalence of severe food insecurity and poor food consumption score in East Hararghe Zone, Oromia, Ethiopia (severe food insecurity n = 4628; poor food consumption score n = 4554).**

20–29 years. 52.24% of households had one or more children under five. 13.40% of the household's WRA respondent were currently pregnant, 87.46% had a history of pregnancy, and 66.31% of WRA respondents were married. 86.02% of WRA respondent's had no formal education and an almost equal percentage of household heads (86.50%) also had no formal education. 28.78% of WRA respondents performed cash-earning work in the past 12 months. Nearly all households, 97.49%, had daily-per-capita food consumption costs <1.61 USD, indicative of food poverty. Only 18.44% used financial services, almost all had land plot access (97.06%), and 22.88% of households raised livestock/cultivated crops with intent to sell. The type of livestock included oxen (6.35%), poultry (19.67%), goats (23.46%), and of note, the far majority of households planted crops they make decisions over (91.17%). Indicators related to WASH were overall low: only 2.36% had hand washing facilities, 7.40% correctly treated drinking water, and 15.85% used improved sanitation facilities (Table 2).

## Severe food insecurity

Households with WRA respondents aged 15–19 years were more likely to experience SFI (AOR = 1.79; 95% CI: 1.36, 2.34) compared to WRA 20–29 years of age. Households with two or more CU5 were less likely to experience SFI (AOR = 0.72; 0.60, 0.88) compared to households with no CU5. Households where the WRA respondent was pregnant had higher odds of SFI (AOR = 1.51; 1.17, 1.96), and the strongest predictor of SFI were households where the WRA respondent had a history of pregnancy (AOR = 3.46; 2.76, 4.33). Households with WRA that performed cash-earning work in the previous year were more likely to experience SFI (AOR=1.35; 1.12, 1.61), and those with daily-per-capita food consumption <1.61 USD was another strong predictor (AOR = 2.98; 1.91, 4.66). The strongest protective associations were households that raised oxen (AOR=0.34; 0.26, 0.45) or raised livestock/cultivated crops to sell (AOR = 0.50; 0.42, 0.60). Households that planted crops had higher odds of SFI (AOR = 1.67; 1.31, 2.13). Although 91% of all households planted crops, high levels of severe food insecurity persisted. Among severely food insecure households, 92.48% planted crops, suggesting this risk factor was driven primarily from by households that planted crops and did not raise livestock. As only 34.98% of severely food insecure households raised target livestock, this supports the observed protective association with raising livestock/cultivating crops to sell. This protective effect is likely attributable to households that either solely raised livestock, or engaged in livestock and crop production for market-oriented purposes. Households with handwashing facilities were 3.83 times more likely to experience SFI (1.92, 7.63), and those that used improved sanitation were 0.61 times less likely to experience SFI (0.50, 0.74) (Table 3) (Fig 2).

**Poor food consumption score.** When compared to WRA 20–29 years, households with the WRA respondent aged 15–19 or 30–49 were less likely to experience poor food consumption (AOR=0.59; 0.46, 0.76 and AOR=0.76; 0.63, 0.91, respectively). Households with two or more CU5 also had lower odds of poor food consumption compared to those with no CU5 (AOR=0.77; 0.64, 0.91), with 52.08% of WRA aged 30–49 years residing in such households. Households with currently pregnant WRA or WRA with a history of pregnancy were less likely to experience poor food consumption (AOR=0.57; 0.47, 0.70; AOR=0.70; 0.55, 0.89, respectively). Although only 11.63% of all WRA with a history of pregnancy were aged 15–19, 62.14% of those WRA aged 15–19 had been pregnant, which may partially explain these parallel protective associations observed. Households in which either the household head or the WRA respondent had no formal education were more likely to have poor food consumption (AOR = 1.46; 1.18, 1.79 and AOR = 1.44; 1.15, 1.81, respectively). Households that planted crops had lower odds of poor food consumption (AOR=0.57; 0.44, 0.75), as did households that saved money (OR=0.32; 0.21, 0.51). In contrast, households that used financial services were more likely to have poor food consumption (AOR = 2.10; 1.69, 2.59). Daily-per-capita food consumption <1.61 USD was the strongest predictor of households with poor food consumption (AOR = 4.01; 2.58, 6.21). Households that raised oxen were less likely to have poor food consumption (AOR=0.68; 0.52, 0.90), and those that raised livestock/cultivated crops to sell had the strongest protective effect (OR=0.40; 0.34, 0.48). Households that used improved sanitation facilities was also associated with poor food consumption (OR=0.57; 0.47, 0.68) (Table 4) (Fig 2).

**Table 3. Factors associated with severe food insecurity from univariate (n = 4,628) and multivariate logistic regression (n = 4,335) among PSNP households in East Hararghe Zone, Oromia, Ethiopia. Crude and adjusted odds ratios displayed.**

| Characteristics | n | Univariate OR | 95%CI | p | Multivariate OR (n = 4,335) | 95%CII | p |
|---|---|---|---|---|---|---|---|
| **Age Group** | 4,628 | | | | | | |
| 20–29 y (Ref) | | 1.00 | . | . | 1.00 | . | . |
| 15–19 y | | 1.33 | (1.06, 1.66) | 0.012 | 1.79 | (1.36, 2.34) | <0.001 |
| 30–49 y | | 1.29 | (1.10, 1.52) | 0.002 | 1.10 | (0.91, 1.34) | 0.314 |
| **Number of CU5 in household** | 4,628 | | | | | | |
| No CU5 in household (Ref) | | 1.00 | . | . | 1.00 | . | . |
| One CU5 in household | | 0.97 | (0.82, 1.15) | 0.749 | 0.88 | (0.73, 1.07) | 0.197 |
| Two or more CU5 in household | | 0.75 | (0.64, 0.89) | 0.001 | 0.72 | (0.60, 0.88) | 0.001 |
| **Demographics** | | | | | | | |
| Currently pregnant (Ref: No) | 4,627 | 1.72 | (1.37, 2.17) | <0.001 | 1.51 | (1.17, 1.96) | 0.002 |
| History of pregnancy (Ref: No) | 4,626 | 2.68 | (2.23, 3.22) | <0.001 | 3.46 | (2.76, 4.33) | <0.001 |
| Married (Ref: No) | 4,628 | 0.91 | (0.78, 1.05) | 0.196 | . | . | . |
| **Education Level** | | | | | | | |
| WRA No school (Ref; formal education) | 4,628 | 0.99 | (0.81, 1.21) | 0.942 | . | . | . |
| Household Head No school (Ref: formal education) | 4,628 | 1.29 | (1.07, 1.57) | 0.009 | 1.12 | (0.89, 1.40) | 0.324 |
| **Economic Activity** | | | | | | | |
| WRA cash earning work last 12 months (Ref: No) | 4,628 | 1.40 | (1.19, 1.64) | <0.001 | 1.35 | (1.12, 1.61) | 0.001 |
| Household saved money (Ref: No) | 4,395 | 0.51 | (0.35, 0.75) | 0.001 | 0.78 | (0.49, 1.23) | 0.284 |
| Daily per capita food consumption costs <1.61 USD (Ref:>=1.61USD) | 4,622 | 2.73 | (1.88, 3.97) | <0.001 | 2.98 | (1.91, 4.66) | <0.001 |
| Used financial services (Ref: No) | 4,398 | 0.67 | (0.56, 0.80) | <0.001 | 0.91 | (0.74, 1.12) | 0.385 |
| Access to a plot of land (Ref: No) | 4,485 | 1.13 | (0.75, 1.69) | 0.551 | . | . | . |
| Household raises/buys livestock/cultivate crops with intent to sell (Ref: No) | 4,396 | 0.49 | (0.42, 0.58) | <0.001 | 0.50 | (0.42, 0.60) | <0.001 |
| **Livestock/Crops** | | | | | | | |
| Household raises oxen (Ref: No) | 4,485 | 0.33 | (0.26, 0.43) | <0.001 | 0.34 | (0.26, 0.45) | <0.001 |
| Household raises poultry (Ref: No) | 4,485 | 1.07 | (0.89, 1.28) | 0.464 | . | . | . |
| Household raises goats (Ref: No) | 4,485 | 0.85 | (0.72, 1.00) | 0.048 | 1.03 | (0.86, 1.23) | 0.781 |
| Household planted crops it made decisions over (Ref: No) | 4,351 | 1.91 | (1.52, 2.39) | <0.001 | 1.67 | (1.31, 2.13) | <0.001 |
| **WASH** | | | | | | | |
| Household has handwashing facilities (Ref: No) | 4,626 | 2.59 | (1.38, 4.85) | 0.003 | 3.83 | (1.92, 7.63) | <0.001 |
| Household correctly treats water (Ref: No) | 4,621 | 1.14 | (0.87, 1.50) | 0.344 | . | . | . |
| Household used improved sanitation facilities (Ref: No) | 4,626 | 0.63 | (0.52, 0.75) | <0.001 | 0.61 | (0.50, 0.74) | <0.001 |

## Discussion

While most relevant research in Ethiopia has assessed the general population, our analysis focuses on a chronically food insecure, drought-affected population. Accordingly, our study revealed high prevalence of severe food insecurity (77.8%) and poor food consumption (69.7%) among PSNP households in East Haraghe Zone. The risk factors for severe food insecurity included households where the WRA respondent was aged 15–19, was currently pregnant, had a history of pregnancy, and performed cash-earning work, along with households that had daily-per-capita food consumption <1.61 USD, planted crops, and had handwashing facilities. Protective factors for severe food insecurity included households that had two or more CU5, raised livestock/crops to sell, raised oxen, and used improved sanitation facilities. Risk factors of poor food consumption included households that had daily-per-capita food consumption <1.61 USD, used financial

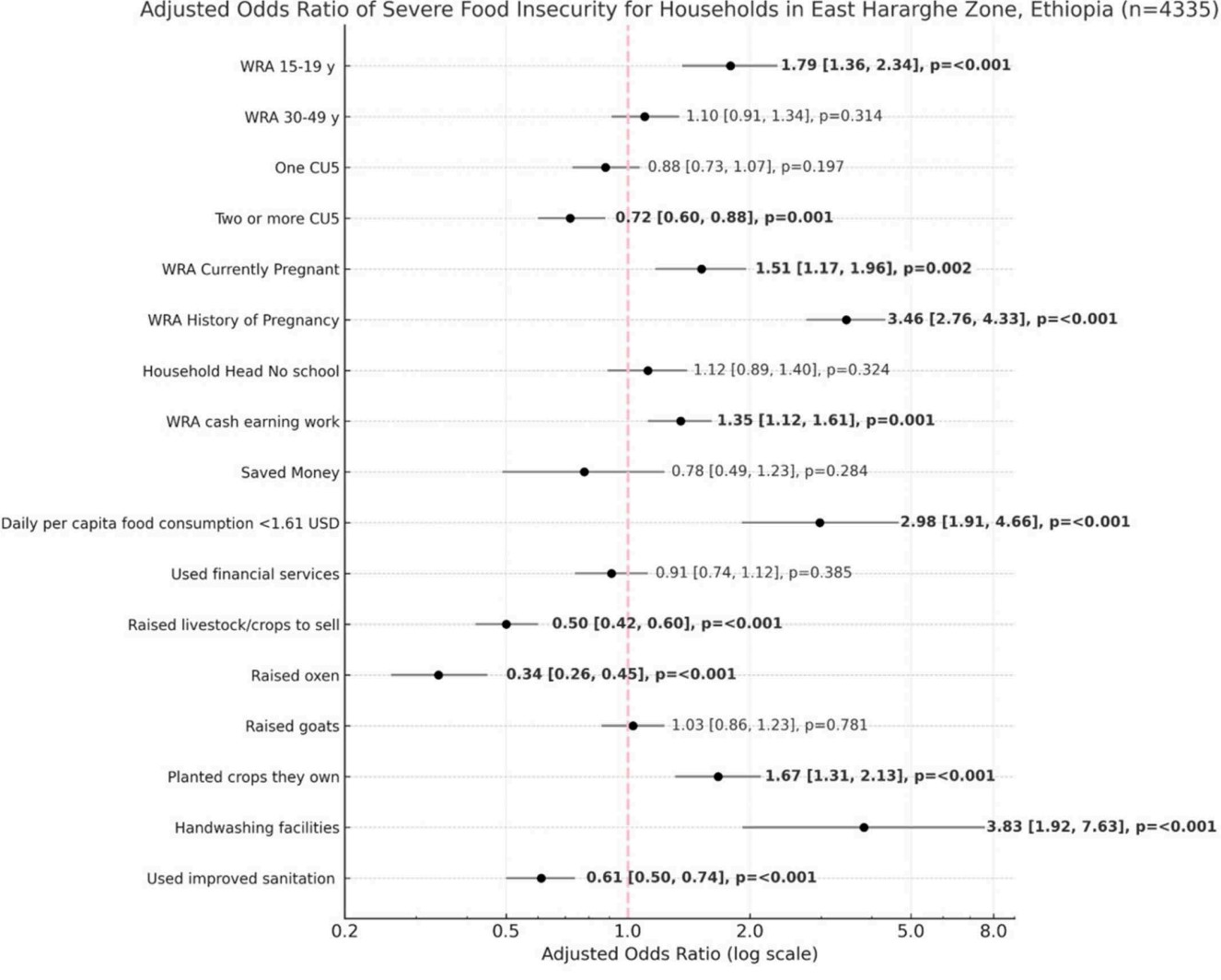

**Fig 2. Results from multivariate logistic regression expressed in a Forest Plot.** Factors associated with severe food insecurity and poor food consumption score at the 95% confidence level (α = 0.05) (Severe Food Insecurity n = 4335; Poor Food Consumption Score n = 4268).

services, and those that had a WRA respondent and/or household head with no formal education. The protective factors for poor food consumption were households where the WRA respondent was aged 15–19, currently pregnant, or had a history of pregnancy, in addition to households that had two or more CU5, saved money, raised livestock/crops to sell, and raised oxen.

The prevalence of food insecurity in Fedis woreda, EHZ was estimated to be 58% by Mulugeta et al [25], while Getaneh et al identified food insecurity prevalence in Rift Valley as 64% [26]. A study by Aweke et al of EHZ smallholder farms identified the prevalence of poor food consumption in the preharvest season as 21.9% [27], while Fite et al in Haramaya detected unacceptable food consumption to be 45.5% [28]. These studies were conducted before intensified drought and record-breaking consecutive seasons of low rainfall [2], which likely contributed to the higher magnitude of severe food security and poor food consumption found in our sample. Additionally, a study by Hiruy et al in southwest Oromia collected data from November to December 2021, deeper into the rainy season that still displayed levels of food

**Table 4. Factors associated with poor food consumption score from univariate (n = 4,554) and multivariate logistic regression (n = 4,268) among PSNP households in East Hararghe Zone, Oromia, Ethiopia. Crude and adjusted odds ratios displayed.**

| Characteristics | n | Univariate OR | 95% CI | p | Multivariate OR (n = 4268) | 95% CI | p |
|---|---|---|---|---|---|---|---|
| **Age Group** | 4,554 | | | | | | |
| 20–29 y (Ref) | | 1.00 | . | . | 1.00 | . | . |
| 15–19 y | | 0.74 | (0.60, 0.90) | 0.003 | 0.59 | (0.46, 0.76) | <0.001 |
| 30–49 y | | 0.90 | (0.77, 1.06) | 0.202 | 0.76 | (0.63, 0.91) | 0.003 |
| **Number of CU5 in household** | 4,554 | | | | | | |
| No CU5 in household (Ref) | | 1.00 | . | . | 1.00 | . | . |
| One CU5 in household | | 0.93 | (0.79, 1.08) | 0.324 | 0.89 | (0.75, 1.06) | 0.177 |
| Two or more CU5 in household | | 0.79 | (0.68, 0.92) | 0.003 | 0.77 | (0.64, 0.91) | 0.003 |
| **Demographics** | | | | | | | |
| Currently Pregnant (Ref: No) | 4,553 | 0.60 | (0.50, 0.72) | <0.001 | 0.57 | (0.47, 0.70) | <0.001 |
| History of pregnancy (Ref: No) | 4,552 | 0.70 | (0.57, 0.86) | 0.001 | 0.70 | (0.55, 0.89) | 0.004 |
| Married (Ref: No) | 4,554 | 1.08 | (0.94, 1.23) | 0.283 | – | – | – |
| **Education Level** | | | | | | | |
| WRA No school (Ref; formal education) | 4554 | 1.62 | (1.36, 1.93) | <0.001 | 1.44 | (1.15, 1.81) | 0.002 |
| Household Head No school (Ref: formal education) | 4554 | 1.54 | (1.29, 1.84) | <0.001 | 1.46 | (1.18, 1.79) | <0.001 |
| **Economic Activity** | | | | | | | |
| WRA cash earning work last 12 months (Ref: No) | 4,554 | 0.78 | (0.68. 0.90) | <0.001 | 0.86 | (0.73, 1.00) | 0.054 |
| Household Saved Money (Ref: No) | 4,326 | 0.40 | (0.28, 0.59) | <0.001 | 0.32 | (0.21, 0.51) | <0.001 |
| Daily per capita food consumption costs <1.61 USD (Ref:>=1.61USD) | 4,548 | 4.89 | (3.30, 7.25) | <0.001 | 4.01 | (2.58, 6.21) | <0.001 |
| Used financial services (Ref: No) | 4,329 | 1.23 | (1.04, 1.46) | 0.019 | 2.10 | (1.69, 2.59) | <0.001 |
| Access to a plot of land (Ref: No) | 4,415 | 0.76 | (0.50, 1.13) | 0.174 | – | -- | – |
| Household raises/buys livestock/crops with intent to sell (Ref: No) | 4,328 | 0.46 | (0.40, 0.54) | <0.001 | 0.40 | (0.34, 0.48) | <0.001 |
| **Livestock/Crops** | | | | | | | |
| Household raises oxen (Ref: No) | 4,415 | 0.60 | (0.47, 0.77) | <0.001 | 0.68 | (0.52, 0.90) | 0.006 |
| Household raises poultry (Ref: No) | 4,415 | 0.77 | (0.66, 0.91) | 0.001 | 0.87 | (0.73, 1.04) | 0.133 |
| Household raises goats (Ref: No) | 4,415 | 0.82 | (0.70, 0.95) | 0.007 | 0.95 | (0.80, 1.12) | 0.519 |
| Household planted crops it made decisions over (Ref: No) | 4,283 | 0.74 | (0.58, 0.95) | 0.017 | 0.57 | (0.44, 0.75) | <0.001 |
| **WASH** | | | | | | | |
| Household has handwashing facilities (Ref: No) | 4,552 | 0.74 | (0.50, 1.10) | 0.139 | . | . | . |
| Household correctly treats water (Ref: No) | 4,547 | 0.94 | (0.74, 1.20) | 0.621 | . | . | . |
| Household used improved sanitation facilities (Ref: No) | 4,552 | 0.59 | (0.50, 0.70) | <0.001 | 0.57 | (0.47, 0.68) | <0.001 |

insecurity at 62.4%, with 28.1% of the sample experiencing severe food insecurity [29]. Another study by Markos et al in southwest Ethiopia noted 14.3% poor FCS [30], while a study of poor households by Dula in Gelan City identified poor FCS to be 58.33% [31]. This emphasizes the importance of focusing academic research on low-income households in drought emergency settings, particularly in highly affected regions such as East Oromia and Somali, as food insecurity and poor food consumption vary depending on the context.

## Sociodemographic characteristics

EHZ is primarily rural, and since households with women that were currently pregnant or had a history of pregnancy were risk factors for SFI, this conflicts with the Areba et al study that detected pregnant women from rural areas were less likely

to be food insecure [32]. Interestingly, while pregnancy and history of pregnancy were risk factors for SFI, these were protective factors for poor FCS. WRA aged 15–19 years exhibited this same pattern. As 62.14% of WRA aged 15–19 had a history of pregnancy, this relationship appears similar in magnitude and directionality with WRA that were pregnant or had a history of pregnancy. This is notable, as a Jebena et al study found female adolescents experienced poorer self-rated health in the context of food insecurity [33]. Contrarily, in our sample, WRA aged 15–19 and 30–49 were protective factors for poor FCS compared to WRA aged 20–29. Households with two or more CU5 did not follow this pattern and were protective factors for SFI and poor FCS, likely mediated by the 52.08% of WRA aged 30–49 that resided in these households. However, since Getacher et al found no association of two or more CU5 on lactating mothers [34], humanitarian and/or community aid may have supported households with CU5 in EHZ.

Our findings suggest the presence of WRA that were aged 15–19, pregnant, or had a history of pregnancy within households may help shape perceptions of food security, despite adequate food consumption. Since FIES provides a perceptual measure of food security, while the consumption-based metric of FCS provides insight on nutrient intake and dietary diversity, it is important to assess concordance between these two measures. Consumption-based measures may fluctuate weekly in food-insecure households, potentially explaining the conflicting results of FCS as a measure of sufficient nutrition [35], while perception of food security status with FIES is based on continual experience over a year [21]. Although these discordant associations may appear paradoxical, the consistent directionality, magnitude, and precision of our estimates reinforce the model's validity. Our findings are also supported by Lain et al, who observed a significant proportion of households in West Africa to experience severe food insecurity according to FIES, despite reporting borderline or adequate FCS [36]. Adequate food consumption over a week may not be reflected in the overall food security status of a household without consistent access, availability, utilization, and stability, or if there are other factors that may change the definition of food security relative to that household. This divergence could be related to a range of factors that may complicate the relationship between food security and food consumption- including but not limited to pregnancy, history of pregnancy, adolescent girls/women, drought, crop-yield, and/or aid programs. Male-headed households could also influence this relationship, as Hiruy et al found female decision-makers were more likely to experience SFI [29]. Future research should collect Middle Upper Arm Circumference in tandem with food insecurity and consumption data to clarify the links between perception, consumption, and nutritional status.

Household heads and WRA with no formal education were risk factors for poor FCS, and this aligns with Markos et al [30] and Sisay et al [37] where household heads with education were predictors of acceptable FCS. According to Hiruy et al [29] and Getacher et al [34], women with formal education were predictors of food security, and while we found no association with SFI, it aligns with our food consumption sample. This indicates that increased education may heighten awareness of food consumption needs, but awareness alone may be insufficient to increase the food security status of households. Since 86% of WRA and 86.5% of household heads did not receive any schooling, this highlights concerns for education in EHZ.

### Economic activity

WRA that performed cash-earning work in the past year was a risk factor for SFI, indicating income-generating activities may be insufficient to enable food security in drought settings. This conflicts with Areba et al, where employed pregnant women were less likely to be food insecure [32], and Getacher et al, where lactating mothers who did not participate in income-generating activities were more likely to be food insecure [34]. As households that saved cash were protective factors for poor FCS, this may indicate that households with the capacity to save cash may allocate it towards food for consumption. But this may not affect the food security status of the household, as it may be unsustainable to save cash during shocks and stressors, where the immediate demand of drought settings could require households to allocate available funds towards food for consumption. It also may be true that WRA could have worked more because of the food insecurity and drought. This potentially explains the statistically significant positive association between cash-earning work

and SFI, and the negative association between cash-earning work and poor FCS bordering on significance (AOR = 0.86; 95% CI = 0.73, 1.00). As FIES measures perception and FCS quantifies consumption, WRA that perceive their household to be food insecure may perform cash-earning work, and conversely those with adequate FCS may not. This finding further highlights the nuances between perceived food security and measured consumption, and additional research is required to clarify the directionality of this relationship.

Since daily-per-capita food consumption costs <1.61 USD was a risk factor for both SFI and poor FCS, and financial service use was a risk factor for poor FCS, households may not have the financial capacity to repay credit while maintaining food security and adequate consumption. Financial services may also be used to alleviate food insecurity, and the directionality of this association requires further research. Since this conflicts with a study by Wubetie et al in Ethiopia, it complicates the impact of financial services such as credit on daily food consumption [38]. However, this furthers the inference that in EHZ limited financial resources are a primary factor that influence both food security and consumption.

### Livestock/crop production

Households that raised livestock and/or cultivated crops to sell and those that raised oxen both displayed distinctly protective associations with SFI and poor FCS. The protective effect of households that raised livestock/crops to sell is likely driven by households that raised either livestock alone, or both livestock and crops to engage with the market, since 91.2% of households planted crops (Table 2). Oxen displayed a highly protective association, but since only 6.35% of our sample raised oxen, this is likely an indicator of households with greater financial assets that facilitate food security and adequate consumption. This is consistent with the assessment by Gebissa et al [39], and the Mulugeta et al study also conducted in EHZ that identified oxen as predictors of food security [25]. While raising livestock, specifically oxen, saw protective associations, households that planted crops they made decisions over were a risk factor for severe food insecurity. This is inconsistent with the Getacher study where home gardens reduce the risk of food security [34], but aligns with the Gebissa study where livestock ownership was a predictor of food security, while land cultivation saw no relationship [40]. This provides further support that livestock is a primary driver of food security, as compared to crop cultivation. However, households in EHZ that planted crops was a protective factor for poor FCS. This indicates planting crops may help improve FCS by increasing availability and accessibility of fruits/vegetables, nuts/seeds, and/or roots/tubers to diversify diets, but may be insufficient to improve food security at the household level in drought settings.

Aweke et al found growing crops and increased livestock and farm income was a protective factor for poor FCS, similar to the protective association in our sample between raising livestock/cultivating crops to sell and planting crops the household made decisions over [27]. Fite et al in Haramaya found women who did not own agricultural land plots were negatively associated with acceptable FCS [28]. Although we found no association with land plot, these results are similar to the protective effect of planting crops on poor FCS [28]. Important staple crops such as tef, maize, and sorghum all decreased during drought conditions in a study by Temam et al., on the implications of drought in dryland Ethiopia [40]. Agricultural production in Oromia has diminished from drought conditions [41], and since planting crops was a risk factor for severe food insecurity, crop production does not appear to be sufficient to maintain food security. Livestock ownership is likely a better facilitator of food security, an inference also ascertained by Getaneh [26]. Sileshi et al found rural households in EHZ with an increased number of livestock owned were negatively associated with poor/borderline FCS, which was used to assess food security [42].This aligns with the protective relationship of raising livestock/crops to sell and raising oxen with both severe food insecurity and poor FCS in our sample. This also aligns with the study by Sisay et al that detected livestock were positively associated with acceptable FCS [37]. A study in northern Ethiopia by Vaitla identified that increased livestock units was positively associated with FCS, similar to the decreased likelihood of poor FCS when the household raised oxen and raised livestock/crops to sell [43]. These results continue to highlight livestock, specifically oxen, as key facilitators of food security and crop production as a tool to supplement food consumption. Optimally, the livestock and crop production of households is sufficient to engage these assets in the market.

However, livestock are not immune to the impact of drought, and this investment holds risk since livestock death could diminish household resilience [5]. Livestock interventions could also benefit from the climate-sensitive approach to agriculture to decrease risk of death. Interventions to improve survivability and maintain the health of livestock is another approach to improve livelihoods and economic capacity to facilitate food security and mitigate shocks/stressors. Since nearly the entire sample planted crops, this is supported by Tofu et al that found pastoralists had lower resilience than agro-pastoralists, which provides further evidence for the protective association of raising livestock and cultivating crops [44]. The protective association displayed by households that raised oxen in our sample and contemporary literature indicates that scaling up oxen breeding may heighten availability, enhance the livestock market, and improve food security and resilience. It has been determined by Getahun et al that with the agricultural uncertainty in Ethiopia, income diversification for farming households through livestock, forest, or non-farm means may be beneficial [45]. The relationship between income diversification such as forest farming and cash crops and its impact on food security should be further explored. Overall, while planting crops may facilitate adequate food consumption, it does not appear sufficient to maintain food security in drought settings, and accumulating alternative assets such as livestock, particularly with a market-oriented approach, demonstrates a strong protective effect on both food security and consumption.

## Water, sanitation and hygiene

Households that used improved sanitation exhibited a strongly protective effect for both SFI and poor FCS. Vaitla et al did not find improved sanitation to be associated with FCS status, it was a predictor of FCS resilience [44]. We suspect households that used improved sanitation is an indicator of households with greater financial resources, a potential explanation for the protective effect. This could also be related to the ease of access to locations suitable for human waste, which could decrease the burden on resources such as time, effort, and money for these households. Although a small sample, households with handwashing facilities were associated with increased severe food insecurity, potentially because these households may have redirected financial resources towards WASH instead of other resources such as food. As the relationship between WASH and food insecurity/consumption has not been thoroughly assessed in literature, our study provides preliminary insight and further research is needed.

## Recommendations

It is important that public health outcome definitions are reassessed as new evidence emerges, as different factors may reveal unique relationships between conceptually distinct, but related measures such as food security and food consumption [46]. The nuances between perception and consumption must be reflected in humanitarian aid and social protection programs by collecting data on both FIES and FCS, as solely relying on FCS may not accurately reflect the food security status of households with adolescent women/girls, pregnant women, and children [36]. Based on our findings, we recommend directing PSNP resources towards households with pregnant/lactating women, mothers, adolescent girls/women, no education, low economic status, and lack of livestock ownership. The relative feasibility of collecting FIES food insecurity data may be an effective surveillance tool to detect households with pregnant women [32], adolescent girls/women, and children that need food aid. Policymakers should consider creating paid PSNP programs to supplement the EHZ school system [37] that leverages the knowledge of experienced farmers to geographically tailor education and enhance awareness of locally available nutrient-dense and climate-resistant crops. We strongly recommend programs that bolster the livestock market and create expanded opportunities by increasing the livestock population through breeding and improving survivability through vaccination [25–27,39,40,43,44]. It can be a useful approach to apply PSNP behavioral change efforts to reinforce the allocation of financial resources towards protein-rich foods and utilize farming to generate supplementary income and diversify diets with pulses, fruits, vegetables, and nuts [46]. We advise integrating food security and WASH programming to diminish diarrheal infections that can be deleterious to nutritional status. Lastly, in the context of the changing financial landscape with the termination of USAID funding, there

must be a coordinated response at the national level that collaborates with local and global agencies to identify and mobilize alternative funding sources.

## Strengths and limitations

This is a cross-sectional study and cannot assess causality. The sample consists of low-income PSNP beneficiaries in Ethiopia that receive government support, which complicates inferences to the general population but enriches literature in globally underserved settings. Other food security measures such as Household Food Insecurity Access Scale may have provided more information on specific aspects of food security, but FIES is tailored to emergency settings. Data was collected by personal recall and may be susceptible to recall bias and social desirability bias. While seasonality is not assessed, food security and food consumption must be analyzed during lean periods of low food availability. Although a large sample size increases the likelihood of significant associations, this is also an indicator of appropriate covariate selection and high statistical power. Missing data was excluded from multivariate regression analyses to allow for complete case analysis and ensure valid comparison. As a result, sample sizes slightly differ by respectively a 6.54% and 6.48% difference between SFI and poor FCS univariate and multivariate models, and a 1.55% difference between the SFI and poor FCS multivariate models.

## Conclusion

The high prevalence of severe food insecurity (77.79%) and poor food consumption (69.74%) indicates a sizeable number of households in EHZ may be nutritionally susceptible. Importantly, our study identified vulnerable PSNP groups such as households with pregnant/lactating women, mothers, adolescent girls/women, no education, low economic status, and no livestock ownership. Notably, households where the WRA respondent was pregnant, had given birth in the past 5 years, was 15–19 years old, and/or planted crops were more likely to experience severe food insecurity, yet less likely to have poor food consumption. Our findings highlight a critical conceptual distinction between the perceptual indicator of food security and quantitative metric of food consumption, that factors primarily related to women/girls pregnancy and age status may influence household perceptions and/or definitions of food security. These findings highlight a critical conceptual distinction- food security is a perceptual, experience-based indicator, whereas food consumption is a quantitative metric influenced by community and environmental factors. Households classified as food insecure may still have adequate food consumption, and conversely adequate food consumption does not necessarily indicate food security. It is critical to recognize this, and collecting data on both can better contextualize household vulnerability and resilience. The variation in predictors of food insecurity and consumption suggest individual definitions of household food security may change depending on the presence of PLW, mothers, adolescent girls/women, and crop production. The strongest predictor of both severe food insecurity and poor food consumption was low daily-per-capita food consumption costs (<1.61 USD), and the strongest protective effects were households that raised livestock/cultivated crops to sell, raised oxen, and used improved sanitation. Households that raised livestock/cultivated crops to sell was the most accurate and precise predictor of food security and adequate food consumption, likely because livestock and crops were used as market assets to improve household financial capacity. Although crop production on its own appeared insufficient to maintain food security, this may facilitate adequate food consumption score through the addition of fruits, vegetables, and nuts/seeds to diets, which may increase the diversity of nutrient intake. It is pivotal to address educational concerns within the zone to increase awareness of the importance of dietary diversity, particularly through self-produced climate-resistant crops that can withstand droughts. Households that used improved sanitation was a strong predictor of food security and adequate consumption, but the low WASH utilization of these food insecure households can exacerbate child malnutrition by exposure to gastrointestinal pathogens that cause nutrient loss. The dual burden of severe food insecurity and poor food consumption poses a threat to current and future generations in EHZ, and it is critical to take data-driven action to progress toward the sustainable development goal of zero hunger in Ethiopia.

## Supporting information

**S1 Table.** *Variable definitions.* This file provides definitions for each variable assessed in this study.
(DOCX)

## Author contributions

**Conceptualization:** Noah Baker, Yunhee Kang, Shannon Doocy.

**Data curation:** Gregory Makabila, Seifu Tadesse.

**Formal analysis:** Noah Baker.

**Funding acquisition:** Gregory Makabila.

**Investigation:** Noah Baker.

**Methodology:** Noah Baker, Yunhee Kang, Shannon Doocy.

**Project administration:** Noah Baker, Gregory Makabila.

**Resources:** Gregory Makabila, Shannon Doocy.

**Supervision:** Yunhee Kang, Shannon Doocy.

**Validation:** Noah Baker, Yunhee Kang.

**Visualization:** Noah Baker.

**Writing – original draft:** Noah Baker.

**Writing – review & editing:** Noah Baker, Yunhee Kang, Gregory Makabila, Seifu Tadesse, Shannon Doocy.

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
