## [Decision Letter · Decision Letter 0]

25 May 2025

PGPH-D-25-00936

Prevalence and risk factors for severe food insecurity and poor food consumption during a drought emergency in Ethiopia

Dear Dr. Kang,

Thank you for submitting your manuscript to PLOS Global Public Health. After careful consideration, we feel that it has merit but does not fully meet PLOS Global Public Health’s publication criteria as it currently stands. Therefore, we invite you to submit a revised version of the manuscript that addresses the points raised during the review process.

While your responses to Reviewer 1's comments about the recent changes to USAID will certainly be of interest to readers, I see this a bit of a "scope creep" from the subject you set out to study.  Please do make sure to update your manuscript in response to the other feedback from Reviewer 1 and as well as the feedback from Reviewer 2, which I think will strengthen this work.

We look forward to receiving your revised manuscript.

Kind regards,

Abraham D. Flaxman, Ph.D.

Academic Editor

Journal Requirements:

1. Please provide additional details regarding participant consent. In the ethics statement in the Methods and online submission information, please ensure that you have specified (1) whether consent was informed and (2) what type you obtained (for instance, written or verbal, and if verbal, how it was documented and witnessed). If your study included minors, state whether you obtained consent from parents or guardians. If the need for consent was waived by the ethics committee, please include this information. If you are reporting a retrospective study of medical records or archived samples, please ensure that you have discussed whether all data were fully anonymized before you accessed them and/or whether the IRB or ethics committee waived the requirement for informed consent. If patients provided informed written consent to have data from their medical records used in research, please include this information. 2. Your current Financial Disclosure states, “The United States Agency for International Development (USAID) Bureau of Humanitarian Assistance (BHA) provided funding for Ifaa Resilience Food Security Activity (Ifaa/RFSA) to a consortium led by Catholic Relief Services in Ethiopia (720BHA21CA00035). Causal Design was contracted for data collection of the baseline survey for the Ifaa/RFSA project, and to develop an Impact Evaluation under the Implementer-Led Evaluation and Learning (IMPEL) Associate Award funded by USAID BHA, which provided the dataset for this study. The funders had no role in study design, analysis, decision to publish, or preparation of the manuscript”. However, your funding information on the submission form indicates that you received funding from “United States Agency for International Development”. Please indicate by return email the full and correct funding information for your study and confirm the order in which funding contributions should appear. Please be sure to indicate whether the funders played any role in the study design, data collection and analysis, decision to publish, or preparation of the manuscript. 3. In the online submission form, you indicated that The dataset used in this study is not publicly available, as it is subject to third-party ownership by Catholic Relief Services. Access may be shared upon reasonable request, pending approval from the authors and Catholic Relief Services. All PLOS journals now require all data underlying the findings described in their manuscript to be freely available to other researchers, either 1. In a public repository, 2. Within the manuscript itself, or 3. Uploaded as supplementary information. This policy applies to all data except where public deposition would breach compliance with the protocol approved by your research ethics board. If your data cannot be made publicly available for ethical or legal reasons (e.g., public availability would compromise patient privacy), please explain your reasons by return email and your exemption request will be escalated to the editor for approval. Your exemption request will be handled independently and will not hold up the peer review process, but will need to be resolved should your manuscript be accepted for publication. One of the Editorial team will then be in touch if there are any issues. 4. Please provide separate figure files in .tif or .eps format. For more information about figure files please see our guidelines:  https://journals.plos.org/globalpublichealth/s/figures https://journals.plos.org/globalpublichealth/s/figures#loc-file-requirements 5. We notice that your supplementary table is included in the manuscript file. Please remove them and upload them with the file type 'Supporting Information'. Please ensure that each Supporting Information file has a legend listed in the manuscript after the references list.

Additional Editor Comments (if provided):

Reviewers' comments:

Reviewer's Responses to Questions

**Comments to the Author**

1. Does this manuscript meet PLOS Global Public Health’s publication criteria ? Is the manuscript technically sound, and do the data support the conclusions? The manuscript must describe methodologically and ethically rigorous research with conclusions that are appropriately drawn based on the data presented.

Reviewer #1: Yes

Reviewer #2: Yes

2. Has the statistical analysis been performed appropriately and rigorously?

Reviewer #1: Yes

Reviewer #2: Yes

3. Have the authors made all data underlying the findings in their manuscript fully available (please refer to the Data Availability Statement at the start of the manuscript PDF file)?

Reviewer #1: Yes

Reviewer #2: No

4. Is the manuscript presented in an intelligible fashion and written in standard English?

Reviewer #1: Yes

Reviewer #2: Yes

5. Review Comments to the Author

Reviewer #1: Dear Author I have 4 comments here:

Tittle - Prevalence and risk factors for severe food insecurity and poor food consumption during a drought emergency in Ethiopia.

Comment- 1, it was an interesting tittle your research aim to evaluate PSNP effectiveness, but currently USIAD and the like funding agency were terminated currently. While, this program highly co-correlated with this external donor therefore how about your recommendation according the current status of this donors?

Comment-2, since your source of data is secondary data how assures the validity and reliability of the data to get actual finding of the outcome variable.

Comment-3, the sample size of you used for food insecurity and food consumption is quite different why this difference happens?

Comment-4, productive safety net program is not much productive effectively even highly supported with donor like USAID and other agencies. In addition USAID currently it was terminated therefore what recommendation you plan for Ethiopian government especially on local intervention that reduce food insecurity.

Reviewer #2: This paper describes food insecurity and food consumption patterns in drought effected areas of Ethiopia. The source of the data is from an NGO, which is why the raw data is not available. I have some thoughts and questions.

I have ordered my comments in alignment with the manuscript with a mixture of major and minor comments.

Introduction

-you use a lot of abbreviations. If they are not reintroduced, don't add them. Specifically line 114 IDP.

-line 117 - why Russia and Ukraine?

- line 118 - are you talking about chronic or acute severe food insecurity?

-line 120 - cholera: again, is it chronic or acute? and how does that impact food insecurity?

-you need more context, and indicator for lines 124-128. Frame the problem, provide prevalence estimates.

-Lines 128-130: I suggest you change the language to say you want to identify a solution

-lines 135 - you mention it here (I wondered why) and once in the discussion that the data were collected during Ramadan or another time of fasting. 1. explain that clearly. 2. how is that related to food insecurity? I can see food consumption and I think that the literature (informal at least) suggest that food consumption patterns change but maybe not quantity. So, that needs to be clarified and contextualized.

Methods

- Please provide additional details on who completed the survey. Was it the primary caregiver who was a WRA? What if there were many WRA in a household. Your results have multiple age groupings for WRA - were they the primary WRA and completed the survey or were they living in the household?

-covariates: more definitions are needed. Financial services, for example, is that using a bank? Later in the results you refer to a loan. That is a big difference and helpful to know up from. Also, your dichotomous variables are not 'opposites'. For example, #5 raised livestock/cultivated crops with intent to sell - the opposite is raised with no intent to sell NOT did not raise. Same with #7. I think it is a definitional issues so please provide more context.

REsults

Tables: If you chose to use the FI sample, you don't need table 2. If you want to show there are no differences, combine them into one table and show statistical testing for each variable showing no difference.

Table - is the variabel household head no school the survey taker or the male head of household? Who is that?

-Line 250: Thiis an example of the WRA question. You state 'households with WRA aged 15-19...' Does that mean the personal who took the survey? As 'primary WRA' or is it a WRA of that age living in the household. Clarify the sentence. If you mean 'respondent' rather than 'household with', please fix.

-I have to say, there were many variables with unexpected findings. Households with more children under the age of 5 had less food insecurity. Also, WRA 15-19 was associated with higher food insecurity but better food consumption. Same with history of pregnancy, pregnancy, and raising crops. I think your discussion touched on some of that but it is really unexpected.

-Table 3 has a ? in 'daily per capita...'

I was unclear about what you were saying in lines 259-261

Discussion:

-starting at 344- what are some reasons?

-Economic activity: I was thinking a lot about directionality, which you did not talk about. It is possible that people worked because they were food insecure, for example. That is true for many of the indicators. Or, they were forced to do income generating activities because of the drought.

-Lines 360+ - I think you are refering to immediate demand versus long term stability - living from "check-to-check" - is that right?

-374 - the use of credit could be because of the food insecurity

-Line 385 - extra word 'above'

-The water, sanitation and hygiene section was really limited. And, your table showed that having handwashing materials was associate with higher food insecurity. What do you think about that?

-lines 464 - I don't know if that is true or not that there are unique features due to fasting. It doesn't align with your argument of long term drought conditions

6. PLOS authors have the option to publish the peer review history of their article (what does this mean? ). If published, this will include your full peer review and any attached files.

**Do you want your identity to be public for this peer review?** For information about this choice, including consent withdrawal, please see our Privacy Policy .

Reviewer #1: No

Reviewer #2: No

---

## [Decision Letter · Decision Letter 1]

1 Aug 2025

Prevalence and risk factors for severe food insecurity and poor food consumption during a drought emergency in Ethiopia

PGPH-D-25-00936R1

Dear Dr. Kang,

We are pleased to inform you that your manuscript 'Prevalence and risk factors for severe food insecurity and poor food consumption during a drought emergency in Ethiopia' has been provisionally accepted for publication in PLOS Global Public Health.

Best regards,

Abraham D. Flaxman, Ph.D.

Academic Editor

Reviewer Comments (if any, and for reference):

Reviewer's Responses to Questions

**Comments to the Author**

1. If the authors have adequately addressed your comments raised in a previous round of review and you feel that this manuscript is now acceptable for publication, you may indicate that here to bypass the “Comments to the Author” section, enter your conflict of interest statement in the “Confidential to Editor” section, and submit your "Accept" recommendation.

Reviewer #1: All comments have been addressed

Reviewer #2: All comments have been addressed

2. Does this manuscript meet PLOS Global Public Health’s publication criteria ? Is the manuscript technically sound, and do the data support the conclusions? The manuscript must describe methodologically and ethically rigorous research with conclusions that are appropriately drawn based on the data presented.

Reviewer #1: Yes

Reviewer #2: Yes

3. Has the statistical analysis been performed appropriately and rigorously?

Reviewer #1: Yes

Reviewer #2: Yes

4. Have the authors made all data underlying the findings in their manuscript fully available (please refer to the Data Availability Statement at the start of the manuscript PDF file)?

Reviewer #1: Yes

Reviewer #2: Yes

5. Is the manuscript presented in an intelligible fashion and written in standard English?

Reviewer #1: Yes

Reviewer #2: Yes

6. Review Comments to the Author

Reviewer #1: Overall aspect of this manuscript so nice. This research asses retrospectively in this case how how about the validation of data to infer general population during a drought emergency of Ethiopia. Since the data source is secondary data how refine or filter the data which was incomplete

Otherwise it was very interesting overall work

Reviewer #2: Thank you for your careful review of my comments and edits. This paper is much stronger and clearer. Well done.

7. PLOS authors have the option to publish the peer review history of their article (what does this mean? ). If published, this will include your full peer review and any attached files.

**Do you want your identity to be public for this peer review?** For information about this choice, including consent withdrawal, please see our Privacy Policy .

Reviewer #1: **Yes: ** Fikremariam Endeshaw

Reviewer #2: No
